# Concurrent Hydrolysis–Fermentation of *Tenebrio molitor* Protein by *Lactobacillus plantarum* KCCM13068P Attenuates Inflammation in RAW 264.7 Macrophages and Constipation in Loperamide-Induced Mice

**DOI:** 10.3390/foods14111886

**Published:** 2025-05-26

**Authors:** Hyun-Sol Jo, Da-Bin Song, Sang-Hee Lee, Kyu-Shik Lee, Jungwoo Yang, Sun-Mee Hong

**Affiliations:** 1Marin Industry Research Institute for East Sea Rim, 22 Haeyanggwahak-gil, Uljin-gun 36315, Gyeongsangbuk-do, Republic of Korea; hyunsoljo@mire.re.kr (H.-S.J.); hhh082828@naver.com (D.-B.S.); 2Korea Food Research Institute, Iseo-myeon, Wanju-gun 55365, Jeollabuk-do, Republic of Korea; shlee@kfri.re.kr; 3Department of Pharmacology, College of Medicine, Dongguk University, 123 Dongdae-ro, Gyeongju 38066, Gyeongsangbuk-do, Republic of Korea; there1@dongguk.ac.kr; 4Department of Microbiology, College of Medicine, Dongguk University, 123 Dongdae-ro, Gyeongju 38066, Gyeongsangbuk-do, Republic of Korea; dbl3jwy@dongguk.ac.kr

**Keywords:** fermented insect protein, *Tenebrio molitor*, *Lactobacillus plantarum*, anti-constipation, bioactive peptides

## Abstract

Constipation is a common gastrointestinal disorder that negatively impacts quality of life and gut function. This study aimed to enhance the functional properties of *Tenebrio molitor* (mealworm) protein hydrolysates through fermentation with *Lactobacillus plantarum* KCCM13068P (TmLp). Compared to non-fermented hydrolysates (HeTm), TmLp exhibited improved protein hydrolysis and increased levels of amino acids associated with intestinal health. In vitro, TmLp showed anti-inflammatory effects in RAW 264.7 macrophages and promoted the growth of beneficial gut bacteria. In a loperamide-induced constipation mouse model, TmLp significantly enhanced fecal output, water content, and intestinal transit. These findings suggest that TmLp may serve as a promising functional food ingredient for relieving constipation and promoting gut health.

## 1. Introduction

Constipation is a prevalent chronic gastrointestinal disorder influenced by various lifestyle factors, such as poor dietary habits, physical inactivity, and psychological stress. This condition not only affects quality of life but can also lead to other physiological disturbances, including gut microbiota imbalances and low-grade intestinal inflammation [1,2]. While pharmacological treatments are commonly used to alleviate constipation, they often carry risks, including dependency, recurrence, and adverse side effects. These challenges highlight the need for safe, natural alternatives that can effectively relieve constipation while promoting overall gut health [1].

In recent years, edible insects have gained attention as promising, next-generation food resources due to their rich nutritional profiles. Insects such as grasshoppers (*Oxya chinensis*), field crickets (*Gryllus bimaculatus*), silkworm pupae (*Bombyx mori*), and mealworm larvae (*Tenebrio molitor*) are particularly abundant in high-quality proteins and essential amino acids like leucine, glutamic acid, and arginine [3,4]. These nutrients are vital in supporting digestive function, modulating inflammatory responses, and maintaining intestinal barrier integrity [5,6,7,8]. However, the digestibility and bioavailability of insect-derived proteins may be limited due to their macromolecular structure, necessitating bioconversion techniques to improve their functional efficacy.

Fermentation, particularly with probiotics, is a well-established method to enhance the nutritional and physiological properties of protein-based foods. This process facilitates the breakdown of high-molecular-weight proteins into smaller peptides and amino acids, thereby increasing their bioavailability [9]. Additionally, fermentation produces bioactive metabolites, such as organic acids, exopolysaccharides, and short peptides, which contribute to gastrointestinal homeostasis and mucosal protection [10]. The selection of the appropriate microbial strain is crucial for optimizing fermentation outcomes. Probiotic species, including *Lactobacillus rhamnosus*, *Lactobacillus acidophilus*, *Lactobacillus casei*, *Lactobacillus reuteri*, *Streptococcus thermophilus*, *Bifidobacterium bifidum*, and *Bifidobacterium longum*, are known to regulate gut microbial levels, modulate the immune system, and reduce inflammation [11].

Among these probiotics, *Lactobacillus plantarum* has garnered significant interest due to its exceptional tolerance for acids and bile, strong adhesion to intestinal epithelial cells, and high survival rate in the gastrointestinal tract [12]. *L. plantarum* is a widely studied probiotic known for its gastrointestinal resilience, mucosal adhesion, and production of beneficial metabolites such as short-chain fatty acids (SCFAs) and antimicrobial peptides. Its anti-inflammatory, antioxidant, and gut motility-enhancing properties make it a strong candidate for improving intestinal health and relieving constipation [12,13]. Furthermore, *L. plantarum* exhibits potent proteolytic activity by producing proteases and peptidases that break down insect-derived macromolecular proteins into smaller peptides and amino acids [13,14]. These bioactive amino acids and peptides, including proline, glutamic acid, and arginine, support nutrient absorption, intestinal motility, and mucosal integrity. Additionally, previous studies have demonstrated that *L. plantarum*-fermented substrates, such as insect-derived protein hydrolysates and dairy proteins, can significantly reduce pro-inflammatory cytokines (e.g., TNF-α, IL-6) in LPS-stimulated RAW 264.7 macrophages and DSS-induced colitis mouse models. These fermented products also promote the growth of beneficial gut microbiota (e.g., Lactobacillus, Bifidobacterium), enhance water retention in the colon, and accelerate intestinal transit, thereby contributing to improved bowel function [15]. Furthermore, they exhibit immunomodulatory effects by suppressing pro-inflammatory cytokines such as TNF-α and IL-6 in LPS-stimulated macrophages and colitis-induced mice [16].

Based on these considerations, this study aimed to evaluate the anti-inflammatory and laxative effects of *L. plantarum*-fermented *T. molitor* protein hydrolysates. To this end, both in vitro and in vivo models were employed to assess cytokine expression, fecal parameters, intestinal transit, and water retention.

## 2. Materials and Methods

### 2.1. Preparation of T. molitor Hydrolysate

*T. molitor* larvae were sourced from a commercial supplier in South Korea (SignalCare Inc., Chungdo-gun, Republic of Korea). The *T. molitor* larvae were defatted to a residual fat content of 3–5% (Wenling Linda Machinery Co., Ltd., Wenling, China), dried using a microwave oven (P70D20TL-D4 Galanz Microwave, Foshan, China) to achieve a moisture content of less than 1–2%, and then pulverized into a fine powder (Wenling Linda Co., Ltd.). A strain of *L. plantarum* KCCM13068P, used for hydrolysis, was originally isolated in 2019 from sediment collected at a fish-farm tank at the National Institute of Fisheries Science (Pohang, Republic of Korea) and deposited in the Korean Culture Center of Microorganisms (KCCM) in 2021. This strain was selected based on previous screening results demonstrating probiotic potential, high viability during fermentation, and the presence of protein hydrolysis-related genes when applied to insect-based substrates. Its gastrointestinal stability and ability to produce bioactive metabolites have also been characterized. Protein hydrolysates were prepared by suspending *T. molitor* powder in distilled water (DW) at a concentration of 4.5% (*w*/*v*). The suspension was autoclaved at 121 °C for 15 min and cooled to room temperature. The cooled suspension was used as the non-cultured *T. molitor* control (HeTm). For fermentation, the *T. molitor* suspension was inoculated with *L. plantarum* KCCM13068P at 1% (*v*/*v*, 1.0 × 10^6^ CFU/mL) and incubated anaerobically at 30 °C for 24 h, resulting in the TmLp sample. Additionally, a separate batch was adjusted to pH 7.5 using 1 N NaOH before fermentation to enhance the efficiency of protein hydrolysis and peptide release. Alkaline conditions have been reported to improve the solubility and enzymatic digestibility of insect-derived proteins, thereby facilitating the production of low-molecular-weight peptides with potential bioactivity [17]. After fermentation, the samples were centrifuged at 3000× *g* (relative centrifugal force, RCF) for 10 min, and the supernatants were freeze-dried to obtain protein hydrolysates. The freeze-dried hydrolysates were stored at −72 °C until further analysis.

### 2.2. SDS-PAGE Analysis

Protein samples were prepared by centrifuging 10% (*w*/*v*) solutions at 8000 rpm for 10 min. The supernatants were collected and subsequently filtered through a 0.45 μm syringe filter. Each filtered sample was mixed with 4× NuPAGE™ LDS sample buffer (Invitrogen, Carlsbad, CA, USA) at a ratio of 1:3 (*v*/*v*) and subjected to SDS-PAGE. For electrophoresis, 20 μL of each prepared sample was loaded onto 15% SDS-polyacrylamide gels and separated in 1× SDS running buffer (25 mM Tris, 192 mM glycine, and 0.1% SDS) at 100 V for 90 min using a PowerPac™ basic power supply (Bio-Rad, Hercules, CA, USA). Protein molecular weight markers (Precision Plus Protein™ Dual Color Standards; Bio-Rad, USA) were used for molecular size estimation. Following electrophoresis, gels were stained with EZ-Gel Staining Solution (Dogenbio, Seoul, Republic of Korea) and rinsed with sterile DW to remove excess dye and enhance band visualization. The gel region corresponding to proteins with weights below 15 kDa was excised and subjected to in-gel digestion for subsequent LC-MS/MS analysis.

### 2.3. Protein Identification by LC-MS/MS

After the excision of gel bands corresponding to proteins smaller than 15 kDa, peptides were extracted and desalted using a C18 solid-phase extraction column (Waters, Milford, MA, USA). Then, the samples were dried using a vacuum concentrator and reconstituted in 0.1% formic acid before LC-MS/MS analysis. Peptide mixtures were analyzed using a NanoUPLC system (Waters, USA) coupled to an LTQ-Orbitrap mass spectrometer (Thermo Electron, Waltham, MA, USA). First, peptides were loaded onto a trap column (Acclaim PepMap 100 C18, 100 μm × 2 cm, 5 μm particle size) and subsequently separated using an analytical column (Acclaim PepMap RSLC C18, 75 μm × 50 cm, 2 μm particle size) with a linear gradient of solvent from 5% to 35% acetonitrile containing 0.1% formic acid at a flow rate of 300 nL/min over 90 min. The mass spectrometer was operated in the positive ion and data-dependent acquisition (DDA) modes. Full MS scans were acquired over an *m*/*z* range of 300–2000 at a resolution of 70,000, followed by higher-energy collisional dissociation (HCD) fragmentation of the top 15 most intense precursor ions at a resolution of 17,500. The normalized collision energy (NCE) was set to 28, and dynamic exclusion was applied for 30 s to prevent repeated fragmentation of the same precursor ions. Raw data files were processed using Proteome Discoverer software (version 2.4, Thermo Fisher Scientific, Waltham, MA, USA) and searched against the NCBI Insect and Human protein databases. Identified peptides with *p* < 0.05 were considered statistically significant.

### 2.4. Amino Acid Composition Analysis

Sample pretreatment for amino acid analysis was conducted according to the Food Code [18,19] (MFDS, 2016). Approximately 1 g of each sample was hydrolyzed with 10 mL of 6 N HCl at 105 °C for 22 h. The hydrolysates were cooled, diluted to 50 mL with 1 N HCl, and further diluted (1 mL in 10 mL) before filtering through a 0.2 μm PTFE membrane filter. Amino acids were analyzed using an ion exchange column (4.6 mm × 60 mm; Hitachi High-Technologies Co., Ltd., Tokyo, Japan) with a UV detector at 570 nm and 440 nm. The mobile phases (PH-1, PH-2, PH-3, PH-4, and PH-RG) and ninhydrin solution were supplied by Mitsubishi Chemical Co. (Tokyo, Japan) and Wako Pure Chemical (Osaka, Japan). All analyses were performed twice, presenting the results as mean values.

### 2.5. Growth-Promoting Effects on Intestinal Beneficial Bacteria

The growth-promoting effects of HeTm and TmLp on *L. plantarum* (KCTC 21024) and *B. bifidum* (KCTC 3202) were assessed. Stock solutions of HeTm and TmLp (1 mg/mL) were prepared in sterile DW and filtered through a 0.45 μm syringe filter, then diluted to concentrations of 0.002, 0.02, 0.25, 0.5, and 1 mg/mL. Phosphate-buffered saline (PBS) served as the control. *L. plantarum* was cultured in MRS broth (MB Cell, Seoul, Republic of Korea) at 30 °C, and *B. bifidum* was cultured in BL broth (MB Cell, Seoul, Republic of Korea) at 37 °C for 24 h each. The bacterial suspensions (5 × 10^5^ CFU/mL) were mixed with the samples at a 1:1 volume ratio (i.e., 100 μL of sample and 100 μL of bacterial suspension) and dispensed into a 96-well plate at a final volume of 200 μL/well and incubated for 24 h. Bacterial growth was measured at 600 nm (SpectraMax, San Jose, CA, USA). All experiments were performed in triplicate.

The growth rate was calculated as follows:Growth rate (%) = (Sample OD_600_)/(Control OD_600_) × 100.

### 2.6. Cytotoxicity Analysis of RAW 264.7 Cells

RAW 264.7 mouse macrophages (ATCC, Manassas, VA, USA) were cultured in DMEM (Gibco, Waltham, MA, USA) with 10% FBS and 100 U/mL penicillin/streptomycin (Gibco, USA) at 37 °C in 5% CO_2_. RAW 264.7 cells were seeded at a density of 1 × 10^5^ cells/mL. A total of 200 μL of the cell suspension (equivalent to 2 × 10^4^ cells/well) was added to each well of a 96-well plate and incubated for 24 h. Media were replaced with fresh media containing HeTm and TmLp at final concentrations of 0, 2, 20, 250, 500, and 1000 μg/mL. After a 24 h incubation, cytotoxicity was evaluated using the EZ-Cytox kit (Dogenbio, Republic of Korea). Absorbance was measured at 540 nm using a SpectraMax ABS microplate reader (Molecular Devices, San Jose, CA, USA), and cell viability was expressed relative to that of the untreated control.

### 2.7. Nitric Oxide (NO) Determination

RAW 264.7 cells were seeded into 6-well plates (Thermo Fisher Scientific, USA) at a density of 8 × 10^3^ cells/mL and incubated at 37 °C with 5% CO_2_ for 24 h. After incubation, the culture medium was removed, and the cells were treated with fresh medium containing lipopolysaccharide (LPS; 1 µg/mL; Sigma-Aldrich, St. Louis, MO, USA) and either HeTm or TmLp (1000 µg/mL) simultaneously. The cells were then incubated for an additional 24 h to assess the anti-inflammatory effects of the samples under LPS-induced inflammatory conditions. PBS and L-NIL (N6-(1-Iminoethyl)-lysine, dihydrochloride in the kit) served as the control and positive control, respectively, and treatment with LPS alone served as the negative control. Following incubation, the cell culture supernatant was collected immediately for NO analysis. The nitric oxide level in the medium was measured using an NO assay kit (Promega, Madison, WI, USA), and absorbance was measured at 540 nm using a SpectraMax ABS microplate reader (Molecular Devices, San Jose, CA, USA).

### 2.8. Quantitative Real-Time PCR (qPCR) Analysis

Total RNA was extracted from cells and analyzed using qPCR for inflammatory gene expression analysis. After treatment, the adherent RAW 264.7 cells were carefully detached using a sterile cell scraper (SPL Life Science, Republic of Korea). The cells were collected by centrifugation at 3000× *g* (RCF) for 5 min at 4 °C and stored at −80 °C until further processing. RNA was extracted using the RNeasy Mini Kit (QIAGEN, Hilde, Germany). RNA concentration and purity were assessed using a NanoDrop spectrophotometer (Thermo Fisher Scientific, Waltham, MA, USA) by measuring absorbance at 260 nm and 280 nm. cDNA was synthesized from 4 μg of total RNA using the cDNA Synthesis Kit (Enzynomics, Daejeon, Republic of Korea) following the manufacturer’s instructions. Gene expression analysis was performed using SYBR Green PCR Master Mix (Enzynomics, Daejeon, Republic of Korea), and the relative expression levels of TNF-α and IL-6 were quantified using qPCR with the following primers [18]: TNF-α (forward: 5′-TAT GGC TCA GGG TCC AAC TC-3′, reverse: 5′-CTC CCT TTG CAG AAC TCA GG-3′); IL-6 (forward: 5′-GGT GAC AAC CAC GGC CTT CCC-3′, reverse: 5′-AAG CCT CCG ACT TGT GAA GTG GT-3′). Primer specificity was confirmed via melt curve analysis, and amplification efficiency was within the acceptable range (90–110%), based on standard curves generated using serial dilutions of cDNA. The qPCR protocol included an initial denaturation at 95 °C for 10 min, followed by 40 cycles of 95 °C for 15 s, 58 °C for 20 s, and 72 °C for 30 s. Gene expression levels were normalized to glyceraldehyde 3-phosphate dehydrogenase (GAPDH; RefSeq: NM_008084.3) using the following primers: forward 5′-TGT GTC CGT TCG TGG ATC TGA-3′ and reverse 5′-CCT GCT TCA CCA CCT TCT TGA-3′. GAPDH was validated as a stable reference gene across all experimental groups using comparative Ct analysis.

### 2.9. Animal Model and Experimental Design

Male Institute of Cancer Research (ICR) mice (6 weeks old) were purchased from KOATECH (Pyeongtaek, Republic of Korea) and housed at room temperature (22 ± 2 °C) with 60 ± 5% relative humidity and a 12 h light/dark cycle. Male mice were specifically selected to minimize hormonal variability associated with the estrous cycle, which may influence gastrointestinal motility and immune responses. This choice aimed to enhance experimental consistency and reduce potential confounding factors. Nevertheless, future studies should include female subjects to assess possible sex-based differences in response to constipation and treatment. The mice had free access to food and water. All animal procedures were approved by the Animal Research Working Committee of Kyung-Sang University (approval number: GNU-250225-M0034; approval date: 25 February 2025). After a 7-day adaptation period, mice were randomly assigned to six groups (n = 5/group): CONT (vehicle control, DW), nCONT (loperamide-only control), HeTm low-dose (1 g/kg), HeTm high-dose (2 g/kg), TmLp low-dose (1 g/kg), and TmLp high-dose (2 g/kg). Randomization was performed using a computer-generated random sequence to minimize selection bias. A sample size of five mice per group was determined based on ethical considerations and previous studies employing similar constipation models, which demonstrated significant outcomes with comparable numbers. Nonetheless, we acknowledge that this sample size may limit statistical power, and future studies incorporating larger cohorts and power analysis are warranted to validate the findings. Furthermore, the absence of a group treated with *L. plantarum* KCCM13068P alone limits the ability to distinguish the effects of the fermented product from those of the probiotic itself. Future studies should include such a control to clarify the underlying mechanisms. TmLp was prepared by fermenting *T. molitor* protein hydrolysates with *L. plantarum* KCCM13068P, followed by lyophilization. The viable bacterial count in the lyophilized TmLp was approximately 3 × 10^9^ CFU/g, as confirmed by serial dilution and plate counting. Both HeTm and TmLp powders were suspended in DW at a concentration of 200 mg/mL and administered orally at 10 mL/kg body weight once daily for 28 days, starting concurrently with the induction of constipation. Loperamide (LPA; 3 mg/kg; Sigma-Aldrich, USA) was administered orally once daily for 10 consecutive days (Days 18–28) to all groups except CONT to induce constipation (Figure 1A). HeTm and TmLp samples were administered from Day 1 to Day 28. Body weight was recorded weekly throughout the 28-day experimental period (Figure 1B).

### 2.10. Measurement of Fecal Parameters and Dietary Intake in Mice

Three parameters were assessed to evaluate the effects of dietary intervention on LPA-induced constipation: dietary intake, fecal output, and fecal water content. Dietary intake and fecal output were measured per cage over 24 h on days 5 and 10 after LPA administration. The measurement time points (Days 5 and 10) were selected based on previous reports indicating significant physiological changes related to constipation and recovery occur during this period in loperamide-induced mouse models [20]. Dietary intake was quantified by measuring the total amount of food consumed per cage, while fecal output was determined by counting the total number of fecal pellets excreted per cage during the same period. For fecal water content analysis, the wet weight of the collected feces was recorded immediately after collection. The samples were then dried at 70 °C for 24 h, and their dry weight was measured. Fecal water content was calculated using the following formula:Fecal water content (%) = [(wet weight of feces − dry weight of feces)]/(wet weight of feces)] × 100.

### 2.11. Measurement of Intestinal Transit and Fecal Retention in Mice

Intestinal transit time was assessed using a 1 mL charcoal meal (3% suspension of activated charcoal in 0.5% (*v/v*) aqueous methylcellulose; Sigma-Aldrich, USA). On the 10th day of LPA administration, mice were fasted for 18 h before being sacrificed by cervical dislocation 30 min after oral administration of the charcoal meal. The distance traveled by the charcoal-labeled contents from the pyloric sphincter to the cecum was measured. Additionally, feces were collected, and the number of charcoal-stained fecal pellets was recorded. The small intestine transit rate was calculated using the following formula:Charcoal transit rate (%) = (distance traveled by activated charcoal)/(total length of the small intestine) × 100.

### 2.12. Data Analysis

All results are expressed as the mean ± standard error of the mean (SEM). Statistical significance was determined using one-way ANOVA conducted in GraphPad Prism 8.0 (GraphPad Software, San Diego, CA, USA). Differences were considered statistically significant at *p* < 0.05.

## 3. Results and Discussion

### 3.1. Characterization of T. molitor Hydrolysate Produced by L. plantarum

SDS-PAGE analysis was performed to compare the protein hydrolysis patterns of HeTm and TmLp. The results revealed distinct differences in protein degradation between the two samples. HeTm, which was not subjected to fermentation, showed minimal protein hydrolysis, with major protein bands observed between approximately 200, 70, 40, and 13 kDa (Figure 2A). These findings suggest that the proteins in HeTm remained largely intact, with only limited degradation. In contrast, TmLp, fermented with *L. plantarum* KCCM13068P, exhibited extensive protein breakdown. The fermentation process led to the complete degradation of high-molecular-weight proteins, resulting in the predominant appearance of protein bands in the 10–15 kDa range (Figure 2A). This indicates that fermentation significantly enhanced protein hydrolysis, breaking down the proteins into smaller peptides with lower molecular weights. The presence of peptides in the 10–15 kDa range in TmLp is particularly notable. Peptides of lower molecular weights are generally associated with improved bioavailability and enhanced functional potential compared to their high-molecular-weight counterparts. These smaller peptides may include bioactive components, such as peptides and amino acids, known to support muscle function, protein metabolism, and overall bioactivity, which could contribute to the observed physiological effects in the subsequent biological assays [21]. These analyses indicate the presence of specific bioactive components and help to determine their contribution to the anti-constipation and anti-inflammatory effects observed in the in vitro and in vivo experiments.

### 3.2. Identification of Bioactive Peptides Using LC-MS/MS

LC-MS/MS analysis was performed to identify bioactive peptides generated from non-fermented (HeTm) and *L. plantarum*-fermented (TmLp) *T. molitor* protein hydrolysates (Table 1). In the HeTm group, four proteins were identified: zinc finger protein 853-like isoform X1 (Insect), laminin subunit gamma-1 precursor, immunoglobulin heavy chain variable region, and VILL protein (Mammalian). These proteins exhibited relatively low detection scores (ranging from 15 to 20), indicating limited hydrolysis or low peptide abundance. This suggests that the release of small, bioactive peptides from insect proteins was low without fermentation. In contrast, the TmLp group showed significantly increased peptide diversity and abundance, with higher identification scores. Notably, insect-derived proteins such as chemosensory protein 5 and mucin-3A-like isoform X1 were detected with high confidence scores (up to 44). Mucin-3A-like isoform X1, a glycoprotein involved in intestinal barrier protection and mucosal lubrication [22], may contribute to improved gut motility and mucosal health when hydrolyzed into bioactive peptides. Additionally, multiple keratin-derived peptides—particularly type I and type II cytoskeletal keratins—were abundant in the TmLp group, with exceptionally high detection scores (up to 746). Keratin peptides are known for their structural stability and potential biological functions, including anti-inflammatory activity and maintenance of epithelial barrier integrity [23]. As illustrated in Figure 2B, a bar graph of the top identified proteins in the TmLp group, keratin, type II cytoskeletal 1, had the highest protein score, indicating its dominant presence and confident identification. Other proteins, such as mucin-3A-like isoform X1 and chemosensory protein 5, were also strongly detected, suggesting their involvement in enhancing gut barrier function and modulating immune responses. These findings indicate that fermentation with *L. plantarum* not only enhances the hydrolysis of macromolecular insect proteins but also generates stable, bioactive peptides that support gastrointestinal health. The identification of mucin- and keratin-derived peptides indicates their potential roles in promoting intestinal motility, strengthening mucosal barriers, and modulating local inflammation. These molecular insights align well with the observed anti-constipation effects in vivo, confirming the functional value of *L. plantarum*-fermented *T. molitor* protein hydrolysates as a promising source of health-promoting peptides.

### 3.3. Amino Acid Composition

To evaluate the impact of lactic acid fermentation on the amino acid composition of *T. molitor* larvae in DW (HeTm) and its fermented counterpart (TmLp), we compared their free and bound amino acid profiles (Table 2). The analysis revealed significant differences in the amino acid concentrations between the two samples. In the free amino acid analysis, HeTm exhibited relatively higher concentrations of essential amino acids, including proline, tyrosine, and alanine. Specifically, proline was quantified at 17.67 mg/100 g in HeTm, while TmLp showed an increase to 162.91 mg/100 g (an increase of 145.24 mg/100 g). Additionally, the bound amino acid analysis revealed that TmLp had significantly higher concentrations of proline (204.55 mg/100 g), glutamic acid (207.13 mg/100 g), and aspartic acid (103.84 mg/100 g) compared to HeTm. When considering the total concentration of amino acids, HeTm contained 56.9 mg/100 g of free amino acids and 98.21 mg/100 g of bound amino acids, resulting in a total of 155.11 mg/100 g. In comparison, TmLp had 405.52 mg/100 g of free amino acids and 1212.38 mg/100 g of bound amino acids, yielding a total of 1617.90 mg/100 g. This represents a 10.4 times increase in total amino acid content in TmLp, indicating that fermentation significantly enriched the amino acid profile. Notably, the total concentrations of proline were higher in TmLp. The quantity of proline was 367.46 mg/100 g in TmLp and 30.57 mg/100 g in HeTm, which differed significantly by 336.89 mg/100 g. While TmLp exhibited a clear increase in total amino acid content compared to HeTm, statistical significance could not be confirmed due to the unavailability of standard deviation data for all samples. Therefore, these values should be interpreted as descriptive trends. Proline supports mucosal barrier integrity and enhances epithelial cell growth and repair, which is essential for maintaining gut health. Additionally, proline functions as an osmoprotectant, helping cells adapt to stress conditions, such as osmotic imbalance or oxidative stress [24]. These properties make proline particularly important in the context of intestinal health, where maintaining epithelial integrity and reducing inflammation are critical to supporting gut function and alleviating disorders such as constipation [25,26]. The increase in proline concentration highlights the role of fermentation in enhancing the amino acid composition of TmLp. Lactic acid bacteria (LAB) can enhance the bioavailability of nutrients by breaking down proteins into bioactive peptides and amino acids, including proline, during fermentation. Furthermore, LAB contribute to maintaining a healthy gut microbiota by producing antimicrobial substances, lowering the gut’s pH, and competitively inhibiting pathogenic bacterial growth. Additionally, LAB have immunomodulatory effects that support mucosal immunity and reduce inflammation. Importantly, fermented products containing LAB and increased levels of specific amino acids, such as proline, may synergistically improve intestinal motility, promote mucosal healing, and help alleviate symptoms of constipation [27,28]. The fermentation of *T. molitor* with *L. plantarum* KCCM13068P not only enhanced overall amino acid content but likely contributed to the elevated levels of bioactive components, such as proline. The combination of LAB activity and increased bioactive amino acids may underlie the observed improvement in gastrointestinal function, suggesting the potential of fermented *T. molitor* hydrolysate as a functional food for constipation relief and intestinal health maintenance. The observed increases in glutamic acid, leucine, and proline following fermentation are consistent with previous reports on fermented insect proteins. For instance, Yi et al. [29] and Zielińska et al. [30] also reported elevated levels of essential and flavor-enhancing amino acids after fermenting mealworms and other edible insects. These changes are attributed to proteolysis and microbial enzymatic activity during fermentation, which improve both nutritional and functional properties. Beyond the effects of specific amino acids, the overall increase and diversification of amino acid composition in TmLp may have created a more favorable intestinal microenvironment. Amino acids not only serve as nutrients for host metabolism but also act as substrates for microbial fermentation, influencing the growth of beneficial bacteria, SCFA production, and luminal pH regulation. These interactions can enhance gut motility, mucosal barrier function, and microbial balance, thereby contributing to constipation relief through both nutritional and microbial pathways [31,32].

### 3.4. Analysis of the Growth-Promoting Effects of HeTm and TmLp on Intestinal Beneficial Bacteria

The growth-promoting effects of HeTm and TmLp on *L. plantarum* and *B. bifidum* were evaluated at various concentrations (0.002, 0.02, 0.25, 0.5, and 1 mg/mL), with PBS used as the negative control (set at 100%). For *L. plantarum* KCTC 21024, HeTm exhibited only a weak growth-promoting effect across all tested concentrations. In contrast, TmLp demonstrated a concentration-dependent increase in bacterial growth, and although the results were not statistically significant, a notable increase of 128.91% was observed at 1 mg/mL (Figure 3). Similarly, following fermentation with *B. bifidum* KCTC 3202, HeTm did not exhibit any growth-promoting effect. However, TmLp significantly enhanced the growth of *B. bifidum*, reaching 109.09% at a concentration of 1 mg/mL (Figure 3). The stronger growth-promoting effect of TmLp on *L. plantarum* may be attributed to species-specific metabolic compatibility, as fermentation with *L. plantarum* likely generated bioactive metabolites—including organic acids, peptides, and oligosaccharides—that preferentially support the growth of the same or closely related probiotic species. Additionally, fermentation appears to have enriched TmLp with functional peptides and amino acids that favor the proliferation of beneficial intestinal bacteria, contributing to a more supportive and balanced microbial environment. These in vitro findings suggest that TmLp has strong potential as a functional ingredient that not only enhances nutrient bioavailability but also selectively promotes the growth of beneficial gut microbiota. Furthermore, the growth-promoting effects observed in vitro are expected to be even more pronounced under in vivo intestinal conditions, where complex interactions with the host environment can amplify microbial proliferation. The particularly robust effect on *L. plantarum* suggests that the fermentation process tailors the metabolic composition of TmLp to enhance symbiosis between the fermented substrate and probiotic strains. This may be pivotal in improving intestinal motility, restoring microbiota balance, and alleviating constipation.

### 3.5. Analysis of Cytotoxicity in RAW 264.7 Cells

The cytotoxic effects of HeTm and TmLp at various concentrations (1.00, 0.50, 0.25, 0.02, and 0.01 mg/mL) on RAW 264.7 macrophage cells were evaluated, as shown in Figure 4A. Neither HeTm nor TmLp exhibited cytotoxicity at any tested concentration, with cell viability consistently maintained at or above 100%. While HeTm-treated cells showed viability levels comparable to those of the negative control, TmLp-treated cells exhibited a significant increase in cell viability, reaching 120.61% at 1.00 mg/mL (*p* < 0.01 vs. untreated control)—representing a 15.70% increase over the control. These findings indicate that TmLp is not only non-cytotoxic but may also promote macrophage proliferation or survival. This effect is likely mediated by bioactive compounds generated during fermentation, which may enhance metabolic or immune-related activity. Since no cytotoxic agent was applied in the experiment, the observed increase in RAW 264.7 cell viability is more plausibly attributed to metabolic or immunological activation rather than cytoprotective effects. Previous studies have demonstrated that fermented peptides and probiotic components can stimulate macrophage proliferation and immune signaling through pathways such as NF-κB and MAPK [33,34]. These results further support the immunomodulatory potential of TmLp as a functional ingredient.

### 3.6. Anti-Inflammatory Effects of HeTm and TmLp

The anti-inflammatory effects of HeTm and TmLp were evaluated in LPS-stimulated RAW 264.7 cells by measuring nitrite production and the expression of pro-inflammatory cytokines TNF-α and IL-6 (Figure 4B–D). LPS treatment significantly increased nitrite levels (4.75 ± 0.25 μM), confirming inflammatory induction. HeTm reduced nitrite levels by 20.4% (3.78 ± 0.06 μM, *p* < 0.05), while TmLp achieved a greater reduction of 51.8% (2.29 ± 0.01 μM, *p* < 0.01 vs. LPS control). TNF-α and IL-6 mRNA levels were elevated 10.5-fold and 3.1-fold, respectively, by LPS. HeTm unexpectedly increased TNF-α expression (17.0-fold), whereas TmLp markedly suppressed TNF-α and IL-6 expression by 6.1-fold and 2.7-fold, respectively. These results demonstrate that TmLp more effectively reduces inflammatory responses than HeTm, likely due to fermentation-enhanced bioactivity. The superior efficacy of TmLp may stem from higher levels of low-molecular-weight peptides generated during fermentation. Di- and tri-peptides, particularly those under 1 kDa, are known to be efficiently absorbed via intestinal transporters such as PepT1, improving systemic bioavailability and biological function [35,36,37].

Similar improvements have been reported in fermented insect proteins. Zielińska et al. [37] observed enhanced antioxidant capacity in fermented Acheta domesticus, and Kim et al. [38] reported immunostimulatory effects in fermented Protaetia brevitarsis. These findings support our results, indicating that fermentation improves both nutritional composition and functional efficacy. The underlying mechanisms may involve fermentation-derived peptides and metabolites that activate immune signaling pathways, such as NF-κB and MAPK [38,39,40].

While the present study did not directly evaluate the molecular mechanisms, previous research suggests that fermented peptides may suppress pro-inflammatory cytokine production via NF-κB inhibition and enhance gut motility through enteric nervous system stimulation. Future studies should explore these pathways in relation to TmLp, including potential roles of toll-like receptors (TLRs), tight junction proteins, and SCFA-mediated G-protein-coupled receptors. These compounds likely contribute to the observed anti-inflammatory and gut health-promoting effects of TmLp.

### 3.7. Body Weight Changes in Mice with Loperamide-Induced Constipation

The effects of HeTm and TmLp on body weight were evaluated in a constipation-induced mouse model (Figure 1A), and body weight gradually increased in all groups throughout the study (Figure 1B). On day 28, the average body weight of the normal group reached 40.12 g, whereas that of the LPA group was 39.54 g, with no significant difference between the two groups. Among the treatment groups, the average body weight of the HeTm high-dose group reached 38.55 g, whereas that of the TmLp high-dose group reached 36.93 g, indicating that the weight loss effect was more pronounced in the TmLp high-dose group compared with that in the HeTm high-dose group. Notably, the high-dose HeTm and TmLp groups were effective in promoting weight loss, with the final body weight in the TmLp high-dose group being 1.25 g lower than that of the HeTm high-dose group and 3.07 g lower than that of the LPA-induced constipation group. These results suggest that both HeTm and TmLp can reduce body weight in constipated mice, with high-dose TmLp demonstrating superior efficacy. This highlights the potential of fermentation to enhance the functional properties of *T. molitor*, suggesting that TmLp intake may help reduce weight gain associated with constipation.

### 3.8. Effects on Fecal Parameters and Dietary Intake in Mice

The effects of HeTm and TmLp on fecal output, excretion efficiency relative to food intake, and fecal water content were evaluated on days 5 and 10 following the initiation of LPA-induced constipation. As shown in Figure 5A, the number of fecal pellets was markedly reduced in the LPA group compared with the normal group, confirming the successful induction of constipation. On day 10, the TmLp low-dose group produced 537 pellets, which was approximately 11% more than the 484 pellets observed in the HeTm low-dose group. Similarly, the TmLp high-dose group produced 453 pellets, reflecting a 3.9% increase from the 436 pellets produced in the HeTm high-dose group, suggesting that TmLp supplementation enhances intestinal motility. Dietary intake remained relatively stable across all groups (Figure 5B), but the fecal pellet-to-food intake ratio was consistently higher in the TmLp-treated groups. On day 5, the TmLp high-dose group had a ratio of 4.46, while that in the Tm high-dose group was 3.46, demonstrating a 28.9% improvement. By day 10, the TmLp low-dose group showed a ratio of 4.13, slightly higher than the 4.00 recorded in the Tm low-dose group, representing a 3.25% increase. This indicates a more efficient excretion process in the TmLp-treated groups. Fecal water content analysis (Figure 5C) revealed that the TmLp high-dose group exhibited 82.03% moisture on day 10, which was a 23% increase compared with that of the HeTm high-dose group (66.65%). Although the TmLp low-dose group showed a moisture content of 63.72%, while that in the HeTm low-dose group was 65.36%, both values were substantially higher than that recorded in the LPA group (39.56%). This indicates that TmLp enhances fecal hydration, further supporting its role in improving bowel function. These findings suggest that TmLp enhances bowel function more effectively than non-fermented HeTm, particularly in terms of fecal output and hydration. The improvement is likely attributed to fermentation-enhanced protein digestibility and the formation of bioactive metabolites, highlighting the potential of TmLp as a functional ingredient for the dietary management of constipation. TmLp not only supports fecal output and hydration but also improves overall gastrointestinal health, making it a promising candidate for managing constipation and promoting digestive health.

### 3.9. Effects of Intestinal Transit and Fecal Retention in Mice

To evaluate the effects of HeTm and TmLp on intestinal motility and fecal retention, the small intestine transit rate and the number of charcoal-stained fecal pellets in the colon were assessed on day 10 following LPA-induced constipation. The results for small intestinal transit and colonic fecal retention are presented in Figure 6A,B. In the charcoal transit assay, the LPA group exhibited a markedly reduced intestinal transit rate of 45.96%, compared to 85.77% in the normal group, confirming the successful induction of constipation. The TmLp high-dose group showed a significant improvement in the transit rate (74.16% vs. 45.96% in LPA group, *p* < 0.01), while the HeTm groups showed no significant difference (*p* > 0.05). The TmLp low-dose group demonstrated a transit rate of 47.55%, suggesting that high-dose TmLp more effectively restored intestinal motility compared to the other treatments. Regarding fecal retention, the LPA group retained an average of 5.80 charcoal-stained fecal pellets in the colon, which was significantly more than the normal group. The HeTm low- and high-dose groups retained 2.20 and 1.80 pellets, respectively. In comparison, the TmLp low-dose group retained 1.40 pellets, which was a 36.4% reduction compared with the HeTm low-dose group, while the TmLp high-dose group retained 1.20 pellets, a 33.3% reduction compared with the HeTm high-dose group. These results indicate that TmLp supplementation more effectively reduces colonic fecal retention than non-fermented HeTm, highlighting its potential to improve intestinal motility and reduce fecal retention in constipation.

The findings from this study demonstrate that fermentation of *T. molitor* with *L. plantarum* KCCM13068P (TmLp) enhances its functional properties, particularly its ability to support intestinal health and alleviate constipation. TmLp significantly improved protein hydrolysis, amino acid composition, and the growth of beneficial gut bacteria, especially *L. plantarum* and *B. bifidum*. In the loperamide-induced constipation mouse model, TmLp also exhibited notable anti-inflammatory activity and promoted intestinal motility. These results suggest that fermented *T. molitor* has strong potential as a functional food ingredient for improving gastrointestinal function, relieving constipation, and enhancing nutrient bioavailability. The fermentation process not only enriched the nutritional profile but also generated bioactive metabolites that may contribute to gut health. However, although several bioactive peptides were identified in TmLp, the present study did not evaluate their stability under gastrointestinal conditions. As peptides are often susceptible to enzymatic degradation during digestion, their in vivo bioavailability and functional activity may differ from in vitro observations. Therefore, attributing the observed anti-inflammatory and laxative effects solely to these peptides should be interpreted with caution. Future studies employing simulated gastrointestinal digestion or in vivo absorption models are warranted to confirm their functional stability and efficacy.

Despite the promising results, this study has several limitations. First, while multiple bioactive peptides were identified, their digestive stability and absorption in vivo were not assessed. Second, the study did not directly investigate the molecular mechanisms underlying the observed effects. Although pathways such as NF-κB and MAPK are suggested based on previous literature, further validation through mechanistic assays is required. Lastly, the experimental design was limited to a single macrophage cell line (RAW 264.7) and one mouse model. Additional studies involving diverse cell types, animal models, and human trials will be necessary to confirm the generalizability of these findings.

## 4. Conclusions

In summary, the fermentation of *T. molitor* protein with *L. plantarum* KCCM13068P significantly enhanced its anti-inflammatory and anti-constipation activities, as shown in both in vitro and in vivo models. These effects were likely mediated by increased production of low-molecular-weight bioactive peptides and beneficial microbial metabolites. The findings highlight the potential of fermented insect protein as a novel functional ingredient for improving gastrointestinal health and immune function. While further mechanistic validation is needed, the study provides a strong foundation for the development of bioactive insect-based nutraceuticals.

## Figures and Tables

**Figure 1 foods-14-01886-f001:**
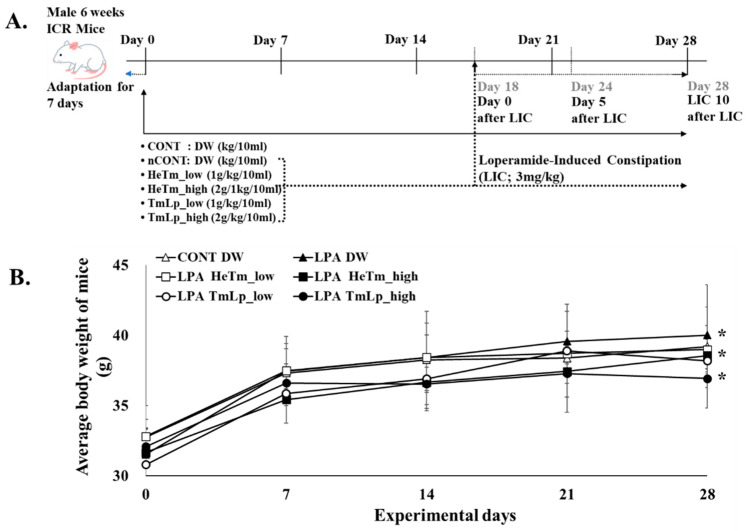
Experimental design and the body weight change in loperamide-induced constipated Institute of Cancer Research (ICR) mice. (**A**) Schematic diagram of the experimental protocol. After a 7-day acclimation period, test samples—non-fermented *T. molitor* hydrolysate (HeTm) and fermented hydrolysate (TmLp)—were administered orally once daily for 28 days. Constipation was induced in all groups except the normal control (CONT) by oral administration of loperamide (LPA; 3 mg/kg) once daily during Days 18 to 28. (**B**) Body weight changes were monitored weekly throughout the 28-day experimental period. Data are presented as the mean ± standard deviation (SD) (n = 5 per group). Asterisks indicate statistically significant differences compared to the control group (* *p* < 0.05).

**Figure 2 foods-14-01886-f002:**
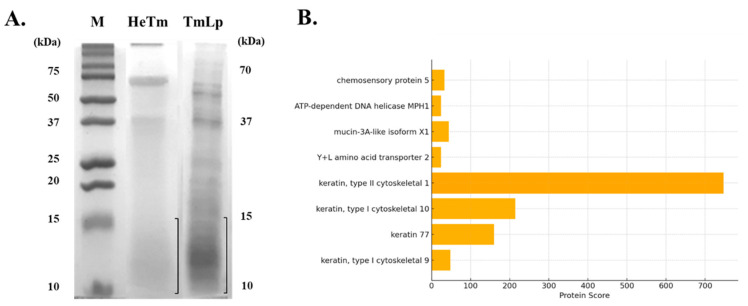
SDS-PAGE and peptide profile of *T. molitor* protein hydrolysates. (**A**) SDS-PAGE analysis was conducted using a 15% gel to compare the protein profiles of HeTm (lane 1) and TmLp (lane 2). M represents the molecular weight protein marker. The boxed region indicates the gel section (<15 kDa) excised for LC-MS/MS analysis. (**B**) Bar graph showing the top peptides identified in the TmLp group based on LC-MS/MS analysis (Table 1). The protein score reflects the confidence and abundance of peptide identification, with keratin, mucin-3A-like, and chemosensory proteins being the most prominent.

**Figure 3 foods-14-01886-f003:**
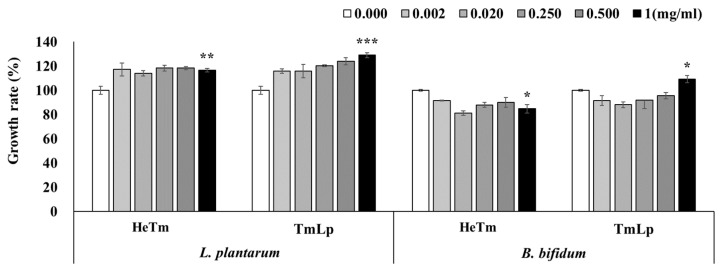
Evaluation of the growth-promoting efficacy of fermented *T. molitor* hydrolysate (TmLp) and non-fermented hydrolysate (HeTm) on beneficial intestinal bacteria, *L. plantarum* (KCTC 21024) and *B. bifidum* (KCTC 3202). Bacterial cultures (5 × 10^5^ CFU/mL) were incubated with varying concentrations (0.002, 0.02, 0.25, 0.5, and 1 mg/mL) of HeTm or TmLp for 24 h. PBS was used as a control. Bacterial growth was measured at 600 nm. Data are presented as the mean ± SD (n = 3). Asterisks indicate statistically significant differences between the experimental groups and the control (* *p* < 0.05; ** *p* < 0.01; *** *p* < 0.001).

**Figure 4 foods-14-01886-f004:**
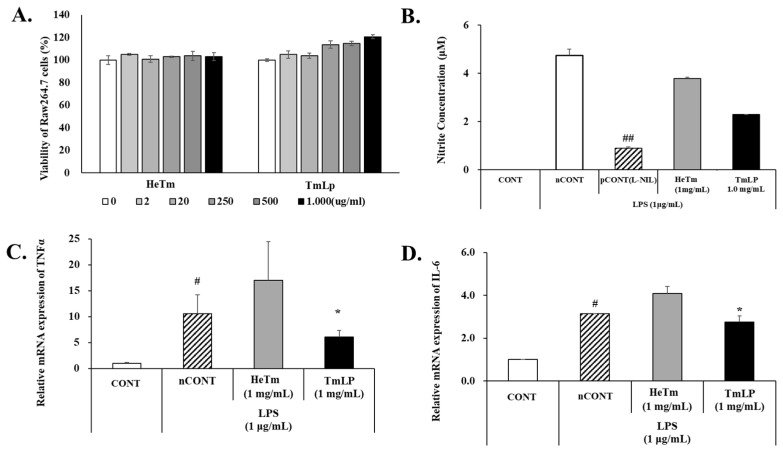
Anti-inflammatory effects of *T. molitor* hydrolysate (HeTm) and fermented *T. molitor* hydrolysate (TmLp) on LPS-stimulated RAW 264.7 cells. RAW 264.7 cells were stimulated with LPS (1 µg/mL) and treated with Tm or TmLp (1000 μg/mL) for 24 h. Cytotoxicity (**A**) and nitric oxide (NO) production (**B**) in the culture supernatant. The expression levels of the pro-inflammatory cytokines TNF-α (**C**) and IL-6 (**D**) were analyzed using qPCR. Data are presented as the mean ± SD (n = 3). Hashtags indicate statistically significant differences between the experimental groups and the control or positive control (# *p* < 0.05, ## *p* < 0.01). Asterisks indicate statistically significant differences between the experimental groups and the LPS-treated negative control (* *p* < 0.05).

**Figure 5 foods-14-01886-f005:**
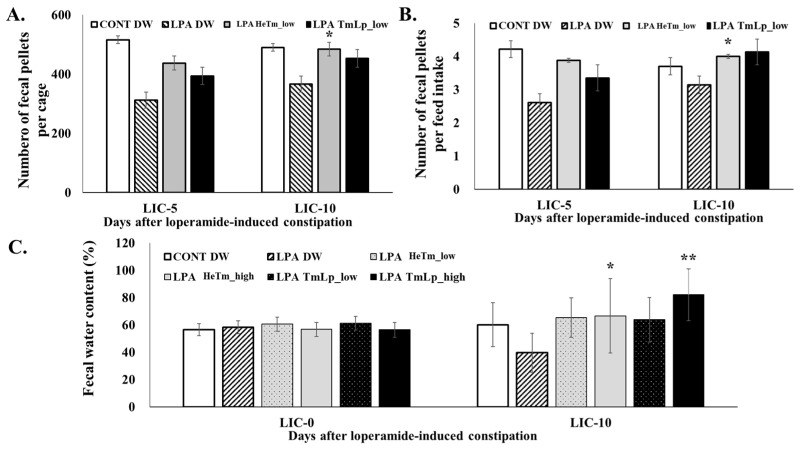
Evaluation of dietary intervention effects on LPA-induced constipation. (**A**) Fecal pellet count per cage (**A**), fecal pellets per unit of feed intake (**B**), and fecal water content (**C**) in experimental mice administered the interventions for 10 consecutive days. LIC-0, 5, and 10 = days after loperamide-induced constipation. Data are presented as the mean ± SD (n = 5 per group). Asterisks indicate statistically significant differences compared to the control group (* *p* < 0.05; ** *p* < 0.01).

**Figure 6 foods-14-01886-f006:**
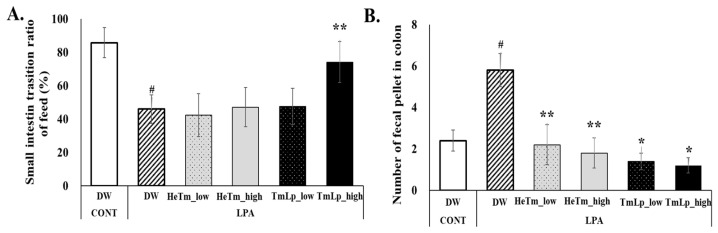
Intestinal transit time assessment and charcoal meal experiment. (**A**) Small intestine transit rate after oral administration of a 3% charcoal meal on ten consecutive days. (**B**) Number of charcoal-stained fecal pellets collected from the colon following the sacrifice of experimental mice. Data are presented as the mean ± SD (n = 5 per group). Hashtags indicate statistically significant differences between the experimental groups and the control and negative control groups (# *p* < 0.05). Asterisks indicate statistically significant differences between the experimental groups and the control and experimental groups (* *p* < 0.05; ** *p* < 0.01).

**Table 1 foods-14-01886-t001:** Identification of peptides in HeTm and TmLp using LC-MS/MS analysis. After SDS-PAGE electrophoresis, the samples were analyzed for proteins with weights smaller than 15 kDa.

	Acc No	Protein Name	Species	Score
HeTm	XP_033230134.1	zinc finger protein 853-like isoform X1	*Belonocnema kinseyi*	15
	NP_002284.3	laminin subunit gamma-1 precursor	*Homo sapiens*	20
	ATO91254.1	immunoglobulin heavy chain variable region	*Homo sapiens*	18
	AAH04300.1	VILL protein	*Homo sapiens*	19
TmLp	UMT69257.1	chemosensory protein 5	*Ophraella communa*	33
	XP_002066172.2	ATP-dependent DNA helicase MPH1	*Drosophila willistoni*	24
	XP_047519532.1	mucin-3A-like isoform X1	*Pieris napi*	44
	XP_002070402.1	Y+L amino acid transporter	*Drosophila willistoni*	24
	UMT69257.1	chemosensory protein 5	*Ophraella communa*	43
	XP_047519532.1	mucin-3A-like isoform X1	*Pieris napi*	39
	NP_006112.3	keratin, type II cytoskeletal 1	*Homo sapiens*	193
	AAG41947.1	keratin 1	*Homo sapiens*	138
	NP_000217.2	keratin, type I cytoskeletal 9	*Homo sapiens*	29
	AAF28956.1	HSPC278, partial	*Homo sapiens*	14
	BAS02858.1	T cell receptor alpha chain V-J-region	*Homo sapiens*	38
	NP_002572.2	pappalysin-1 preproprotein	*Homo sapiens*	18
	NP_006112.3	keratin, type II cytoskeletal 1	*Homo sapiens*	746
	AFA52006.1	keratin 1	*Homo sapiens*	225
	NP_000412.4	keratin, type I cytoskeletal 10 isoform 1	*Homo sapiens*	214
	aai22559.1	Keratin 77	*Homo sapiens*	160
	BAS02858.1	T cell receptor alpha chain V-J-region	*Homo sapiens*	76
	NP_000217.2	keratin, type I cytoskeletal 9	*Homo sapiens*	48

**Table 2 foods-14-01886-t002:** Amino acid components of HeTm and TmLp. Values are expressed as mean ± SD (estimated). Statistical comparisons were not conducted.

Amino Acid (mg/100 g)		HeTm		TmLp	
		Free Amino Acids	Bound Amino Acids	Free Amino Acids	Bound Amino Acids
Essential	Threonine (T)	0.67 ± 0.10	4.30 ± 0.65	8.49 ± 1.02	48.99 ± 5.88
	Valine (V)	3.15 ± 0.47	7.18 ± 1.08	23.68 ± 2.84	64.25 ± 7.71
	Methionine (M)	-	1.99 ± 0.29	0.35 ± 0.04	14.36 ± 1.72
	Isoleucine (I)	1.83 ± 0.27	3.21 ± 0.48	12.99 ± 1.56	40.27 ± 4.89
	Leucine (L)	1.10 ± 0.16	4.28 ± 0.64	12.04 ± 0.91	58.79 ± 7.05
	Phenylalanine (F)	0.70 ± 0.11	4.15 ± 1.87	6.90 ± 0.73	33.56 ± 4.02
	Histidine (H)	4.33 ± 0.65	6.00 ± 0.90	26.68 ± 4.01	51.98 ± 6.24
	Lysine (K)	1.22 ± 0.18	4.22 ± 0.63	12.04 ± 1.44	69.78 ± 8.37
	Tryptophan (W)	1.95 ± 0.29		16.73 ± 2.01	-
Conditionally Essential	Arginine (R)	6.78 ± 1.02	7.11 ± 1.07	-	57.17 ± 6.86
	Cysteine (C)			-	-
	Glycine (G)	0.94 ± 0.10	3.78 ± 0.57	7.97 ± 0.96	67.44 ± 8.09
	Proline (P)	17.67 ± 2.65	12.90 ± 1.94	162.91 ± 19.55	204.55 ± 24.55
	Tyrosine (Y)	5.08 ± 0.76	5.19 ± 0.78	48.28 ± 5.79	48.28 ± 5.79
Non-Essential	Aspartic acid (D)	0.51 ± 0.08	5.30 ± 0.79	2.98 ± 0.36	103.84 ± 12.46
	Serine (S)	0.59 ± 0.09	5.06 ± 0.76	4.94 ± 0.59	54.77 ± 6.57
	Glutamic acid (E)	3.60 ± 0.54	18.77 ± 2.82	21.88 ± 2.63	207.13 ± 24.86
	Alanine (A)	6.78 ± 1.02	4.77 ± 0.72	36.66 ± 4.39	87.22 ± 10.47
Total A.A		56.9 ± 8.53	98.21 ± 4.91	405.52 ± 20.28	1212.38 ± 60.62

## Data Availability

The data presented in this study are available upon request from the corresponding author due to privacy restrictions.

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
