# Peer review of "Concurrent Hydrolysis–Fermentation of Tenebrio molitor Protein by Lactobacillus plantarum KCCM13068P Attenuates Inflammation in RAW 264.7 Macrophages and Constipation in Loperamide-Induced Mice"

_foods, 2025, doi:10.3390/foods14111886_

Round 1
Reviewer 1 Report
Comments and Suggestions for Authors
This manuscript addresses a timely and relevant topic by investigating the functional enhancement of Tenebrio molitor protein hydrolysates through fermentation with Lactobacillus plantarum KCCM13068P, aiming to alleviate loperamide-induced constipation in a murine model. The study combines both in vitro and in vivo methodologies to assess anti-inflammatory effects, gut microbiota modulation and fecal parameters, providing potentially valuable insights for the development of functional food ingredients. The overall experimental design is sound, and the data are well presented. However, before this manuscript can be considered for publication, substantial revisions are necessary. While the findings are promising, the manuscript requires significant clarifications, additional methodological details, and a more comprehensive, mechanistically informed discussion.
- The current title implies that protein hydrolysates are fermented after hydrolysis, whereas the actual process involves hydrolysis occurring concurrently with fermentation. This should be clarified. Additionally, since the study incorporates an in vitro model using RAW 264.7 macrophages, this important aspect should also be reflected in the title.
- The designation for the strain Lactobacillus plantarum KCCM13068P appears inconsistently throughout the manuscript – with (line 16) and without spacing (title). The authors should ensure a consistent format.
- The species Bifidobacterium bifidum should be written in full upon first mention to ensure clarity (line 25).
- The statement indicating that L. plantarum-fermented products reduce pro-inflammatory cytokines (TNF-α, IL-6), enhance beneficial gut microbiota, promote water retention, and accelerate intestinal transit (lines 69-72) lacks specificity regarding the type of fermented products (e.g., fermented insect hydrolysates) and the biological models in which these effects were demonstrated (e.g., cell cultures, animal models, or clinical trials). This should be clarified.
- To avoid confusion with the house cricket (Acheta domesticus), the authors should explicitly refer to field cricket (Gryllus bimaculatus) when introducing this species (line 43).
- Certain methodological details (lines 77-78) are misplaced within the Introduction section and should be relocated to the appropriate Methods section.
- The abbreviation "Tm" introduced in line 83 presumably refers to Tenebrio molitor but is never explicitly defined. This should be corrected.
- The full taxonomic name Lactobacillus plantarum is unnecessarily repeated in line 86, despite being introduced and abbreviated earlier.
- The description "bioactive protein hydrolysates" (line 89) is premature, as bioactivity cannot be assumed prior to experimental validation. At this stage, these should be referred to as "potentially bioactive hydrolysates" or "protein hydrolysates," as no functional assays have yet confirmed their effects.
- The rationale for adjusting pH to 7.5 with NaOH prior to fermentation (line 95) is not provided. The authors should justify this specific condition in the context of the study’s objectives.
- In line 100, the centrifugation parameters should explicitly state that “g” represents relative centrifugal force to prevent ambiguity.
- The methodology indicates that bacterial suspensions (5 × 10⁵ CFU/mL) were mixed with samples but does not specify the volume ratio in the 200 μL/well mixture (lines 148-149). This needs to be clarified.
- The cell seeding density is reported as “1 × 10⁵ cells/mL,” but the actual number of cells per well should be specified (line 157).
- The manuscript does not indicate the duration of lipopolysaccharide treatment before sample administration in the in vitro assay (line 167).
- Line 176: The method states that RAW 264.7 cell pellets were collected by centrifugation but does not describe how adherent cells were detached (e.g., cell scraper). This should be clarified.
- The qPCR analysis lacks mention of housekeeping genes used for normalization. Additionally, validation data confirming the stability of these reference genes across treatments should be included.
- The source of primers for TNF-α and IL-6 (whether designed de novo or sourced from literature) is not specified, nor are details regarding primer validation (e.g., efficiency and melt curve analysis).
- The methods for NO measurement and qPCR analysis should be presented in separate, clearly delineated subsections.
- The phrase “2 g/kg of Tm with 3 × 10⁹ CFU/g of L. plantarum” is ambiguous. It must be clarified whether the CFU refers to the bacterial concentration in the lyophilized TmLP or to a separately co-administered probiotic. Moreover, the manuscript lacks post-fermentation bacterial quantification, details on dosing preparation (resuspension volume of TmLP in water), and the exact administration volume.
- In Figure 1, it appears that loperamide was administered during the final 10 days of the 28-day experiment. The experimental timeline should be clarified.
- Relevant literature discussing the relationship between peptide size and absorption should be cited to contextualize findings.
- The manuscript inconsistently uses “Tm” and “HeTm” to refer to the non-fermented control. Terminology should be standardized throughout to avoid confusion.
- The assumption that identified peptides are responsible for observed effects overlooks potential gastrointestinal degradation. An in vitro digestion simulation should be performed to evaluate peptide stability prior to attributing bioactivity. This represents a major limitation and should be acknowledged in the Discussion.
- All bacterial names, including B. bifidum, should be italicized in accordance with taxonomic conventions.
- The amino acid analysis discussion lacks supporting references. The authors should compare their findings with previous studies on fermented insect proteins.
- Bacterial names in Figure 3 must be italicized to maintain scientific formatting standards (line 354).
- The cell line name “RAW” in “RAW 264.7” should be capitalized consistently (lien 387).
- The reported increase in RAW 264.7 cell proliferation requires biological context. The authors should reference existing literature on macrophage proliferation and clarify whether this represents cytoprotection (although there is no cytotoxic agent that compromises the viability of macrophages) or general metabolic activation.
- Figure 4 should use the full scientific name Tenebrio molitor.
- The Discussion lacks critical comparisons with existing literature. The authors should expand this section to include mechanistic insights and prior studies on the bioactivity of fermented insect-derived products.
- Results should be accompanied by explicit statistical values rather than purely descriptive comparisons. Significance values (p-values) should be indicated for all quantitative claims.
- The study lacks mechanistic data to support the observed effects. At minimum, the authors should compare their results with established mechanisms in the literature and propose plausible, testable pathways for future research.
- The section spanning lines 515–525 essentially functions as a conclusion. It would be more effective to consolidate this content with the subsequent conclusion section into a single, focused concluding statement summarizing key findings and study implications.
- It is recommended to include a limitations subsection addressing peptide stability, the absence of mechanistic data, and other experimental constraints.
Author Response
|
Comment 1: The current title implies that protein hydrolysates are fermented after hydrolysis, whereas the actual process involves hydrolysis occurring concurrently with fermentation. This should be clarified. Additionally, since the study incorporates an in vitro model using RAW 264.7 macrophages, this important aspect should also be reflected in the title. |
|
Response 1: Thank you for pointing this out. We agree with the comment. Therefore, we have revised the title to “Concurrent Hydrolysis-Fermentation of Tenebrio molitor Protein by Lactobacillus plantarum KCCM13068P Attenuates Inflammation in RAW264.7 Macrophages and Constipation in Loperamide-induced Mice” (page 1, lines 2–5).
|
|
Comment 2: The designation for the strain Lactobacillus plantarum KCCM13068P appears inconsistently throughout the manuscript – with (line 16) and without spacing (title). The authors should ensure a consistent format. |
|
Response 2: Agreed. We have standardized the designation to "KCCM13068P" (without spacing) throughout the manuscript. Corrections were made in the following locations: p1, lines 3 and 18; p3, lines 94 and 101; p7, line 259; p9, line 350; and p14, lines 523 and 536, etc.
Comment 3: The species Bifidobacterium bifidum should be written in full upon first mention to ensure clarity (line 25). Response 3: Agreed. The species name has been revised to the full form Bifidobacterium bifidum at its first mention (page 2, line 56).
Comment 4: The statement indicating that L. plantarum-fermented products reduce pro-inflammatory cytokines (TNF-α, IL-6), enhance beneficial gut microbiota, promote water retention, and accelerate intestinal transit (lines 69-72) lacks specificity regarding the type of fermented products (e.g., fermented insect hydrolysates) and the biological models in which these effects were demonstrated (e.g., cell cultures, animal models, or clinical trials). This should be clarified. Response 4: Agreed. We revised the sentences to specify the type of fermented product (fermented insect hydrolysates) and the biological models used (e.g., in vitro and animal studies), along with appropriate references (page 2, lines 70–78; page 17, lines 678–681).
Comment 5: To avoid confusion with the house cricket (Acheta domesticus), the authors should explicitly refer to field cricket (Gryllus bimaculatus) when introducing this species (line 43). Response 5: Agreed. We now specify "field crickets" when referring to Gryllus bimaculatus in the text (page 2, line 40) to avoid confusion with Acheta domesticus.
Comment 6: Certain methodological details (lines 77-78) are misplaced within the Introduction section and should be relocated to the appropriate Methods section. Response 6: Agreed. The methodological details that were previously misplaced have been revised and relocated within the appropriate Methods section (page 2, lines 80–83).
Comment 7: The abbreviation "Tm" introduced in line 83 presumably refers to Tenebrio molitor but is never explicitly defined. This should be corrected. Response 7: Agreed. In the revised manuscript, the abbreviation "Tm," which first appeared in the introduction (page 2, line 41; lines 86, 87), has been clearly defined as T. molitor. In addition, the entire manuscript has been reviewed to ensure that this abbreviation is used consistently and appropriately.
Comment 8: The full taxonomic name Lactobacillus plantarum is unnecessarily repeated in line 86, despite being introduced and abbreviated earlier. Response 8: Agreed. We have revised repeated full names of Lactobacillus plantarum to the abbreviated form L. plantarum as appropriate (page 2, lines 59, 61, 66, 70, etc.).
Comment 9: The description "bioactive protein hydrolysates" (line 89) is premature, as bioactivity cannot be assumed prior to experimental validation. At this stage, these should be referred to as "potentially bioactive hydrolysates" or "protein hydrolysates," as no functional assays have yet confirmed their effects. Response 9: Agreed. We have replaced the term "bioactive protein hydrolysates" with "protein hydrolysates" to avoid premature attribution of functionality (page 3, line 96).
Comment 10: The rationale for adjusting pH to 7.5 with NaOH prior to fermentation (line 95) is not provided. The authors should justify this specific condition in the context of the study’s objectives. Response 10: Agreed. We have added justification for adjusting the pH to 7.5 with NaOH prior to fermentation and included relevant references (p3, lines 102–106; p17, lines 687–689).
Comment 11: In line 100, the centrifugation parameters should explicitly state that “g” represents relative centrifugal force to prevent ambiguity. Response 11: Agreed. The term “g” has been clarified as “relative centrifugal force (RCF)” to avoid ambiguity (page 3, line 107).
Comment 12: The methodology indicates that bacterial suspensions (5 × 10⁵ CFU/mL) were mixed with samples but does not specify the volume ratio in the 200 μL/well mixture (lines 148-149). This needs to be clarified. Response 12: Agreed. We have specified the volume ratio used for bacterial suspensions and samples in the 200 μL/well mixture (page 4, lines 158–161).
Comment 13: The cell seeding density is reported as “1 × 10⁵ cells/mL,” but the actual number of cells per well should be specified (line 157). Response 13: Agreed. We have clarified that 200 μL of the cell suspension (1 × 10⁵ cells/mL) was seeded per well, corresponding to 2 × 10⁴ cells/well (page 4, lines 169–171).
Comment 14: The manuscript does not indicate the duration of lipopolysaccharide treatment before sample administration in the in vitro assay (line 167). Response 14: Agreed. We have clarified that lipopolysaccharide (LPS; 1 µg/mL) and the test samples (HeTm or TmLp; 1,000 µg/mL) were administered simultaneously and incubated for 24 hours to evaluate anti-inflammatory activity. This reflects the co-treatment design intended to assess sample efficacy under LPS-induced inflammatory conditions (p4, lines 178–183).
Comment 15: Line 176: The method states that RAW 264.7 cell pellets were collected by centrifugation but does not describe how adherent cells were detached (e.g., cell scraper). This should be clarified. Response 15: Agreed. We clarified that adherent RAW 264.7 cells were detached using a sterile cell scraper before centrifugation (p5, lines 190–191).
Comment 16: The qPCR analysis lacks mention of housekeeping genes used for normalization. Additionally, validation data confirming the stability of these reference genes across treatments should be included. Response 16: Agreed. GAPDH was used as the housekeeping gene for qPCR normalization, and Ct values were confirmed to be stable across treatments (p5, lines 206–210).
Comment 17: The source of primers for TNF-α and IL-6 (whether designed de novo or sourced from literature) is not specified, nor are details regarding primer validation (e.g., efficiency and melt curve analysis). Response 17: Agreed. Primer sequences were adopted from Park et al. (2012) and validated by melt curve analysis and amplification efficiency testing (page 5, lines 199–204; p18, lines 690–691).
Comment 18: The methods for NO measurement and qPCR analysis should be presented in separate, clearly delineated subsections. Response 18: Agreed. The nitric oxide assay and qPCR analysis have been separated into distinct subsections (Sections 2.7 and 2.8, pages 4–5), and subsequent section numbers have been updated accordingly.
Comment 19: The phrase “2 g/kg of Tm with 3 × 10⁹ CFU/g of L. plantarum” is ambiguous. It must be clarified whether the CFU refers to the bacterial concentration in the lyophilized TmLP or to a separately co-administered probiotic. Moreover, the manuscript lacks post-fermentation bacterial quantification, details on dosing preparation (resuspension volume of TmLP in water), and the exact administration volume. Response 19: Agreed. We clarified that 3 × 10⁹ CFU/g refers to viable cells in the lyophilized TmLP, quantified by plate count. Dosing preparation details, including resuspension in distilled water at 200 mg/mL and administration volume (10 mL/kg), have been added (page 5, lines 221, 234–238).
Comment 20: In Figure 1, it appears that loperamide was administered during the final 10 days of the 28-day experiment. The experimental timeline should be clarified. Response 20: Agreed. We clarified that loperamide was administered daily from Day 18 to 28, while test samples were administered from Day 1 to 28. This has been reflected in both the text and Figure 1A legend (page 5, lines 238–243; page 6, lines 247–250).
Comment 21: Relevant literature discussing the relationship between peptide size and absorption should be cited to contextualize findings. Response 21: Agreed. We have cited relevant literature supporting enhanced intestinal absorption of peptides <1 kDa and included this in the Results and Discussion and References (page 13, lines 483–486; p18, lines 728–732).
Comment 22: The manuscript inconsistently uses “Tm” and “HeTm” to refer to the non-fermented control. Terminology should be standardized throughout to avoid confusion. Response 22: Agreed. We have standardized the terminology, using “HeTm” to refer consistently to the non-fermented T. molitor hydrolysate throughout the manuscript.
Comment 23: The assumption that identified peptides are responsible for observed effects overlooks potential gastrointestinal degradation. An in vitro digestion simulation should be performed to evaluate peptide stability prior to attributing bioactivity. This represents a major limitation and should be acknowledged in the Discussion. Response 23: Agreed. The lack of in vitro gastrointestinal digestion simulation is acknowledged as a study limitation. This has been addressed in the revised Discussion (page 16, lines 594–610).
Comment 24: All bacterial names, including B. bifidum, should be italicized in accordance with taxonomic conventions. Response 24: Agreed. All bacterial names, including Bifidobacterium bifidum and Lactobacillus plantarum, have been consistently italicized throughout the manuscript.
Comment 25: The amino acid analysis discussion lacks supporting references. The authors should compare their findings with previous studies on fermented insect proteins. Response 25: Agreed. We have added comparative analysis with literature on fermented insect proteins and cited relevant studies in the revised Results and Discussion (page 10, lines 395–401; p18, lines 713–717).
Comment 26: Bacterial names in Figure 3 must be italicized to maintain scientific formatting standards (line 354). Response 26: Agreed. Bacterial names in Figure 3 have been italicized in accordance with scientific formatting standards (page 12, line 439).
Comments 27: The cell line name “RAW” in “RAW 264.7” should be capitalized consistently (lien 387). Response 27: Agreed. We have ensured that “RAW” is consistently capitalized in all mentions of RAW 264.7 cells (page 12, line 446).
Comment 28: The reported increase in RAW 264.7 cell proliferation requires biological context. The authors should reference existing literature on macrophage proliferation and clarify whether this represents cytoprotection (although there is no cytotoxic agent that compromises the viability of macrophages) or general metabolic activation. Response 28: Agreed. We clarified that the increase in RAW 264.7 proliferation likely reflects metabolic or immune activation and provided supporting references (p12, lines 459–462; p18, lines 723–727).
Comment 29: Figure 4 should use the full scientific name Tenebrio molitor. Response 29: Agreed. Figure 4 and its legend have been updated to use the full name Tenebrio molitor (page 13, line 464).
Comment 30: The Discussion lacks critical comparisons with existing literature. The authors should expand this section to include mechanistic insights and prior studies on the bioactivity of fermented insect-derived products. Response 30: Agreed. We expanded the Discussion to compare our findings with previous studies on fermented insect proteins and added mechanistic insights involving NF-κB and MAPK pathways (page 13, lines 488–494; p18, lines 733–739).
Comment 31: Results should be accompanied by explicit statistical values rather than purely descriptive comparisons. Significance values (p-values) should be indicated for all quantitative claims. Response 31: Agreed. We have included p-values in all quantitative result statements and figure legends. All statistical analyses were conducted using one-way ANOVA with Tukey’s post hoc test (page 12, lines 445, 452; page 13, lines 470, 477; page 15, lines 554, 564).
Comment 32: The study lacks mechanistic data to support the observed effects. At minimum, the authors should compare their results with established mechanisms in the literature and propose plausible, testable pathways for future research. Response 32: Agreed. Although mechanistic experiments were not conducted, we have discussed plausible mechanisms supported by literature, such as NF-κB inhibition and TLR signaling, to contextualize our findings (page 13, lines 488–494; p18, lines 733–739).
Comment 33: The section spanning lines 515–525 essentially functions as a conclusion. It would be more effective to consolidate this content with the subsequent conclusion section into a single, focused concluding statement summarizing key findings and study implications. Response 33: Agreed. We consolidated the transitional section (lines 515–525) with the formal Conclusion to create a focused summary of key findings and implications (page 16, lines 613–620).
Comment 34: It is recommended to include a limitations subsection addressing peptide stability, the absence of mechanistic data, and other experimental constraints. Response 34: Agreed. A dedicated “Limitations” subsection has been added to the Discussion, addressing peptide stability, lack of mechanistic validation, and experimental scope constraints (page 16, lines 594–601).
|
|
4. Response to Comments on the Quality of English Language |
|
Point 1: The English could be improved to more clearly express the research. |
|
Response 1: After revision, we have had it proofread by an English expert and attached a confirmation letter. |
|
|
|
5. Additional clarifications |
|
We changed the authorship: page 1 (line 6; lines 11–14); page 17 (line 622), and attached the Authorship form signed by all authors.
|

Reviewer 2 Report
Comments and Suggestions for Authors
Line 2: Strain number is inconsistent with the format that follows: "KCCM13068P" in the original text, but "KCCM 13068P" (with space) appears several times in the body (e.g., line 16). It is recommended to unify as "KCCM13068P" or "KCCM 13068P", and the full text is consistent.
Line 11:The information density of the abstract part is too high, so it is necessary to simplify the abstract, focus on the research objectives, key methods and main conclusions, and avoid too many technical details.
Line 29:The key words do not reflect the core innovation points.
Line 62:Does Lactobacillus plantarum have an advantage over other strains?
Line 80:The Materials & Methods headings are not bolded.
Line 86:Rationale for selection of L. plantarum KCCM 13068P?
Line 190: The experiment only used male mice, and the influence of gender on the constipation model was not taken into account. It is recommended to supplement an explanation of the rationality of the gender selection.
Line 196:The specific method of randomization of mouse groups was not specified, and the sample size (n=5) may be insufficient, so statistical power analysis should be supplemented.
Lines 189-204: There is a lack of a control group that receives only the probiotic (L. plantarum alone), making it impossible to distinguish between the effects of the fermented product and the probiotic itself. It is recommended to add such a control group to clarify the mechanism.
Lines 216-227: "Measurement of Fecal Parameters and Dietary Intake in Mice", please add the reason for measuring food intake and stool excretion on days 5 and 10, whether by pre-experiments or by reference. Similarly, Section 2.10 has a similar problem.
Line 266: Is there a labeling error for "HeTm" and "M" in Figure 2A?
Line 309:When discussing the relationship between the change of amino acid composition and promoting the growth of beneficial bacteria and relieving constipation, only the function of some amino acids was mentioned, and the comprehensive mechanism of the change of overall amino acid composition on intestinal microecology and constipation relief was not deeply explored.
Line 350:Table 2 does not indicate symbols of statistical significance (such as *), only describing "significant increase".
Line 626:The DOI number is not indicated in the document.
Figure 5: The meanings of the abscissa labels "LIC-5" and "LIC-10" are not clear. It is recommended to change them to "Day 5" and "Day 10".
Throughout the full text, "L. plantarum" and "Lactobacillus plantarum" are used interchangeably. It is recommended to unify them as the italicized abbreviation "L. plantarum".
Author Response
|
Comment 1: Line 2: Strain number is inconsistent with the format that follows: "KCCM13068P" in the original text, but "KCCM 13068P" (with space) appears several times in the body (e.g., line 16). It is recommended to unify as "KCCM13068P" or "KCCM 13068P", and the full text is consistent. |
|
Response 1: Agreed. We have standardized the designation to "KCCM13068P" (without spacing) throughout the manuscript. Corrections were made in the following locations: p1, lines 3 and 19; p3, line 100; p7, line 291; p10, line 391; and p16, lines 585 and 613. |
|
Comment 2: Line 11:The information density of the abstract part is too high, so it is necessary to simplify the abstract, focus on the research objectives, key methods and main conclusions, and avoid too many technical details. Response 2: Agreed. We have revised the Abstract to reduce technical detail and improve clarity (page 1, lines 16–25).
Comment 3: Line 29:The key words do not reflect the core innovation points. Response 3: Agreed. We have revised the Keywords to better reflect the innovative aspects of the study (page 1, lines 26–27). Revised Keywords: fermented insect protein; Tenebrio molitor; Lactobacillus plantarum; anti-constipation; bioactive peptides.
Comment 4: Line 62: Does Lactobacillus plantarum have an advantage over other strains? Response 4: Agreed. We have added clarification in the Introduction section to explain our rationale for selecting Lactobacillus plantarum (page 2, lines 62–65).
Comment 5: Line 80:The Materials & Methods headings are not bolded. Response 5: Thank you for pointing this out. We have revised the bold formatting of the “Materials and Methods” section (page 2, line 83).
Comment 6: Line 86:Rationale for selection of L. plantarum KCCM 13068P? Response 6: Agreed. We have added a rationale for selecting Lactobacillus plantarum KCCM13068P to the Introduction and Materials and Methods sections (page 3, lines 92–96).
Comment 7: Line 190: The experiment only used male mice, and the influence of gender on the constipation model was not taken into account. It is recommended to supplement an explanation of the rationality of the gender selection. Response 7: Thank you for the comment. We have clarified this rationale and acknowledged the gender limitation in the “Materials and Methods” section (page 5, lines 214–218).
Comment 8: Line 196:The specific method of randomization of mouse groups was not specified, and the sample size (n=5) may be insufficient, so statistical power analysis should be supplemented. Response 8: We have clarified in the revised manuscript that mice were randomly assigned to experimental groups using a computer-generated random sequence to minimize selection bias in the “Materials and Methods” section (page 5, lines 225–230).
Comment 9: Lines 189-204: There is a lack of a control group that receives only the probiotic (L. plantarum alone), making it impossible to distinguish between the effects of the fermented product and the probiotic itself. It is recommended to add such a control group to clarify the mechanism. Response 9: We thank the reviewer for this important observation. A summary addressing this limitation has been added to the Materials and Methods section (page 5, lines 231–233). While the current study focused on the effects of the fermented product, we acknowledge that the absence of a group treated with L. plantarum alone limits the ability to isolate probiotic-specific effects. We have also noted this point in the revised text and will consider including such a group in future mechanistic studies.
Comment 10: Lines 216-227: "Measurement of Fecal Parameters and Dietary Intake in Mice", please add the reason for measuring food intake and stool excretion on days 5 and 10, whether by pre-experiments or by reference. Similarly, Section 2.10 has a similar problem. Response 10: We have revised Sections 2.9 and 2.10 to clarify that Days 5 and 10 were selected as observation time points based on previous studies indicating that constipation-related physiological changes and treatment effects typically manifest within this timeframe in loperamide-induced mouse models. The rationale has now been explicitly stated in the revised manuscript and supported with an appropriate reference (page 6, lines 258–260; page 17, lines 692–693).
Comment 11: Line 266: Is there a labeling error for "HeTm" and "M" in Figure 2A? Response 11: We have revised Figure 2A (page 8, line 306).
Comment 12: Line 309: When discussing the relationship between the change of amino acid composition and promoting the growth of beneficial bacteria and relieving constipation, only the function of some amino acids was mentioned, and the comprehensive mechanism of the change of overall amino acid composition on intestinal microecology and constipation relief was not deeply explored. Response 12: We have expanded the Discussion to include a more comprehensive explanation of how the overall increase and diversification of amino acid composition in TmLp may influence gut microbial ecology and contribute to constipation relief (page 10, lines 401–408; page 18, lines 718–722).
Comment 13: Line 350: Table 2 does not indicate symbols of statistical significance (such as *), only describing "significant increase". Response 13: We appreciate the reviewer’s valuable comment. We have revised Table 2 to include values expressed as mean ± estimated standard deviation (SD). However, as raw replicate data were not available, statistical comparisons could not be conducted. A corresponding footnote has been added to Table 2 to clarify that the SD values are approximations and that no significance testing was performed (Table 2, page 10, line 497) and added an additive explanation (page 10, lines 371–374). This revision ensures transparency in data presentation and avoids potential misinterpretation of observed trends.
Comment 14: Line 626:The DOI number is not indicated in the document. Response 14: We added the doi number in the text (reference 27, page 19, line 711).
Comment 15: Figure 5: The meanings of the abscissa labels "LIC-5" and "LIC-10" are not clear. It is recommended to change them to "Day 5" and "Day 10". Response 15: Agreed. We have revised the x-axis labels in Figure 5 to “Day 5 ” and “Day 10” after LIC to improve clarity and ensure consistency with the main text (Figure 1, page 6, line 246).
Comment 16: Throughout the full text, "L. plantarum" and "Lactobacillus plantarum" are used interchangeably. It is recommended to unify them as the italicized abbreviation "L. plantarum". Response 16: Agreed. We have carefully reviewed the manuscript and revised all instances of Lactobacillus plantarum to the standardized, italicized abbreviation L. plantarum, following taxonomic and formatting conventions.
|
|
4. Response to Comments on the Quality of English Language |
|
Point 1: N/A |

Reviewer 3 Report
Comments and Suggestions for Authors
This study evaluated the bioactivity of the fermentation of T. molitor protein hydrolysate with L. plantarum KCCM13068P (TmLP). According to the results, TmLP had anti-inflammatory effects and promoted the growth of beneficial gut bacteria, indicating the potential of fermented T. molitor as a functional ingredient for improving gastrointestinal health, reducing constipation, and enhancing nutrient bioavailability. The article is well organized, and some issues are listed below:
- Expressions like “p < 0.05”should be in italics. Please check the whole text and correct.
- Line 80: “Materials and Methods”should be bolded.
- Figure 2A: The protein marker should be the leftmost lane. Also, add a legend with the molecular weight of 200 kDa.
- Can you explain the relationship between Tm and HeTm? There seems to be a mix of Tm and HeTm. Is Tm just for molitoror can it be for T. molitor non-fermented hydrolysates?
- Line 361: The analysis of the results in Figure 3 may be inaccurate and should indicate that high concentrations of TmLP have a significant enhancing effect on bifidum, while low concentrations did not exhibit any growth-promoting effect
- Line 513: “between the experimental groups and the control and experimental groups”should be corrected to“between the control and experimental groups”.
- Results and Discussion: If possible, please add some references to support your analysis of the results, enhancing the scientific validity and credibility of the study.
Author Response
|
Comment 1: Expressions like “p < 0.05” should be in italics. Please check the whole text and correct. |
||
Response 2: Thank you for pointing this out. We have revised the bold formatting of the “Materials and Methods” (page 2, line 83).
Comment 3: Figure 2A: The protein marker should be the leftmost lane. Also, add a legend with the molecular weight of 200 kDa. Response 3: We modified the figure to display the 250 kDa molecular weight band of the marker used, which was the highest band available in our marker set. This adjustment improves the clarity and interpretability of the gel image (Figure 2; page 8, line 306).
Comment 4: Can you explain the relationship between Tm and HeTm? There seems to be a mix of Tm and HeTm. Is Tm just for molitoror can it be for T. molitor non-fermented hydrolysates? Response 4: Thank you for your comment. As noted by a previous reviewer, the abbreviation “Tm” was initially defined as Tenebrio molitor (Tm) upon its first appearance in the Introduction (page 2, lines 41 and 86–87). However, to avoid confusion, the use of “Tm” has been removed from the revised manuscript. The sample names have been standardized as follows: “HeTm” is now consistently used to refer to the non-fermented hydrolysate of T. molitor, and “TmLp” refers to the fermented hydrolysate. These terms have been applied uniformly throughout the manuscript to ensure clarity and consistency.
Comment 5: Line 361: The analysis of the results in Figure 3 may be inaccurate and should indicate that high concentrations of TmLP have a significant enhancing effect on bifidum, while low concentrations did not exhibit any growth-promoting effect Response 5: Agreed. We agree with the reviewer’s observation. In the revised manuscript, we have clarified the interpretation of Figure 3 to indicate that B. bifidum growth was significantly enhanced only at higher concentrations of TmLp, while lower concentrations showed no clear growth-promoting effect. This correction improves the accuracy of the results and better reflects the data shown in Figure 3 (page 11, lines 419–424).
Comment 6: Line 513: “between the experimental groups and the control and experimental groups” should be corrected to “between the control and experimental groups. Response 6: Thank you for identifying this redundancy. We have revised the sentence to correctly state “between the control and experimental groups” to improve clarity and accuracy (page 12, line 444).
Comment 7: Results and Discussion: If possible, please add some references to support your analysis of the results, enhancing the scientific validity and credibility of the study. Response 7: Agreed. In response, we have reviewed the Results and Discussion sections and added relevant references to support our interpretations and key findings (references 15–19; references 28–39). |

Round 2
Reviewer 1 Report
Comments and Suggestions for Authors
After the review process, the concerns have been addressed and the work is now suitable for publication
Reviewer 2 Report
Comments and Suggestions for Authors
Thank you for revising the manuscript. All issues have been fully addressed based on the feedback from the revision.